# A Phase 2 Clinical Trial on the Use of Cibinetide for the Treatment of Diabetic Macular Edema

**DOI:** 10.3390/jcm9072225

**Published:** 2020-07-14

**Authors:** Noemi Lois, Evie Gardner, Margaret McFarland, David Armstrong, Christine McNally, Nuala Jane Lavery, Christina Campbell, Rita I Kirk, Daiva Bajorunas, Ann Dunne, Anthony Cerami, Michael Brines

**Affiliations:** 1The Wellcome-Wolfson Institute for Experimental Medicine, Queens University Belfast, 97 Lisburn Road, Belfast BT9 7BL, UK; David.Armstrong@belfasttrust.hscni.net (D.A.); nualajane.lavery@belfasttrust.hscni.net (N.J.L.); 2The Northern Ireland Clinical Trials Unit (NICTU), 1st Floor, Elliott Dynes Centre, Royal Hospitals, Belfast BT12 6BA, UK; evie.gardner@nictu.hscni.net (E.G.); christine.mcnally@nictu.hscni.net (C.M.); christinax.campbell@nictu.hscni.net (C.C.); 3Pharmacy Department, Royal Group of Hospitals, Belfast BT12 6BA, UK; Margaret.McFarland@belfasttrust.hscni.net; 4Araim Pharmaceuticals, 580 White Plains Road, Suite 210, Tarrytown, NY 10591, USA; rkirk@araimpharma.com (R.I.K.); dbajorunas@araimpharma.com (D.B.); adunne@araimpharma.com (A.D.); acerami@araimpharma.com (A.C.); mbrines@araimpharma.com (M.B.)

**Keywords:** ARA 290, cibinetide, diabetes, diabetic macular edema, diabetic retinopathy, DME, erythropoietin, helix B surface peptide

## Abstract

Purpose: Evaluating the effects of cibinetide in diabetic macular edema (DME). Methods: Phase 2 trial. Naïve patients with >400 µm central retinal thickness (CRT) DME in one/both eyes were recruited (May 2016–April 2017) at the Belfast Health and Social Care Trust. The study eye was that with best vision and lowest CRT. Patients self-administered cibinetide 4 mg/day subcutaneously for 12 weeks. Primary and secondary outcomes: mean change from baseline to week 12 in best corrected visual acuity (BCVA), CRT, central retinal sensitivity, tear production, patient-reported outcomes, adverse events and antibodies to cibinetide. Descriptive statistics were used; exploratory analyses focused on non-study eyes, diabetic control, serum cytokines and albuminuria. Results: Nine patients were recruited; eight completed the study. There was no improvement in mean change baseline-week 12 in BCVA (−2.9 + 5.0), CRT (10 + 94.6 microns), central retinal sensitivity (−0.53 + 1.9 dB) or tear production (−0.13 + 7.7 mm), but there was an improvement in National Eye Institute Visual Function Questionnaire (NEI VFQ-25) composite scores (2.7 + 3.1). Some participants experienced improvements in CRT, tear production, diabetic control and albuminuria. No serious adverse events/reactions or anti-cibinetide antibodies were seen. Conclusions: The cibinetide 12-week course was safe. Improvements in NEI VFQ-25 scores, CRT, tear production, diabetic control and albuminuria, observed in some participants, warrant further investigation. Trial Registration: EudraCT number: 2015-001940-12. ISRCTN16962255—registration date 25.06.15.

## 1. Introduction

Diabetic macular edema (DME) is a leading cause of sight loss in people with diabetes with an estimated overall age-standardised prevalence of 6.8% [1]. DME can be treated with laser, anti-vascular endothelial growth factor (anti-VEGF) therapies or steroids. Randomised clinical trials (RCTs) have shown efficacy of anti-VEGFs; however, their route (intravitreal) and frequency (monthly initially and still frequently thereafter) of administration, requirement of long-term treatment (40–50% of patients require injections at 4–5 years following initiation) and of additional therapies to control DME (~50% of patients on anti-VEGFs still require laser treatment), potential complications (increased intraocular pressure, cataract, retinal detachment, endophthalmitis) and cost are important shortcomings [2,3,4]. Furthermore, the efficacy of anti-VEGFs observed in RCTs is not matched to that in real-world clinical practice [2]. Thus, the search for other therapies must continue.

The pathogenesis of DME is complex. Inflammation and increased vascular permeability are recognised inter-related events that lead to DME (reviewed by Stitt et al.) [5]. Early in the disease process there are interactions between circulating immune cells (e.g., monocytes, neutrophils) and retinal vasculature beginning with leukostasis, leading to vessel occlusion, breakdown of the blood retinal barrier (BRB) and subsequent edema [6]. Neutrophils and monocytes/macrophages are major sources of pro-inflammatory cytokines, including VEGF [7].

Over the past 20 years, an endogenous system of tissue protection and repair has been identified and characterised; it utilises erythropoietin (EPO), locally produced in the setting of cellular stress. EPO is the endogenous ligand for a heteromeric cytokine receptor comprised of EPO receptor and CD-131 subunits, the innate repair receptor (IRR), which is also upregulated by stressors [8]. Abundant data show the high relevance of this paracrine/autocrine system of tissue protection and repair for diabetic retinopathy (DR), with impressive efficacy of EPO, as well as engineered mimetics, in preclinical models. IRR signalling activates an anti-inflammatory-reparative response, particularly from macrophage/microglial populations, endothelial cells, and neurons. These activities include the powerful attenuation of tissue-damaging pro-inflammatory cytokines, preservation of the BRB, and prevention of neuronal and endothelial cell apoptosis, among others [8]. 

Cibinetide, also known as ARA 290 and helix B surface peptide (HBSP), is a synthetic 11 amino acid peptide, derived from the structure of the B helix of EPO, with marked anti-apoptotic, anti-inflammatory and anti-permeability effects, recapitulating the effects of the endogenous EPO [9,10,11]. Unlike EPO, is not haematopoietic and, thus, is free of the possible side effects of EPO, which can be life threatening (reviewed by Reid and Lois) [12]. In an experimental model of DR, cibinetide administered systemically inhibited vascular leakage and edema [10] and protected against retinal blood vessel and neuroglial degeneration [11]. Additionally, cibinetide led to improvements in metabolic control in both preclinical [13,14] and clinical [15] studies. Hence, it is possible that this peptide could have a beneficial effect on the treatment of DR and DME and warrants investigation. 

## 2. Experimental Section

This was a prospective, interventional, exploratory, investigator-led, pilot, phase 2, open-label clinical trial to determine whether cibinetide, administered at a daily dose of 4 mg subcutaneously for 12 weeks, could have beneficial effects in patients with DME. 

The trial was conducted in accordance with the ethical principles of the Declaration of Helsinki. Ethical approval (ORECNI ref No.: 15/NI/0197) and Clinical Trial Authorisation from the Medicines and Healthcare products Regulatory Agency (MHRA) (Ref No.: 32485/0029/001-0001) were obtained. The trial was registered with the European Union Drug Regulating Authorities Clinical Trials (EudraCT) (Ref No.: 2015-001940-12), the International Standard Randomised Controlled Trial Number (ISRCTN) (Ref No.: ISRCTN16962255) and clinicaltrials.gov. 

### 2.1. Eligibility Criteria 

Patients with type 1 or 2 diabetes and DME were eligible if they had center-involving DME with central subfield retinal thickness (CRT) of >400 µm, in one or both eyes, as determined using spectral domain optical coherence tomography (SD-OCT), were >18 years of age with clear media and naïve to previous treatments for DME. Patients were excluded if they had edema related to other causes, active proliferative diabetic retinopathy requiring treatment or other eye diseases that could affect vision, hazy media, allergy to fluorescein, had received previous treatment for DME, had CRT of <400µm in the study eye, were receiving systemic/topical steroids, erythropoiesis stimulating agents or investigational medications. Pregnant or breastfeeding women, pre-menopausal women unwilling to take a pregnancy test prior to trial entry or use contraception during the study and men who had female partners and were unwilling to undertake adequate precautions to prevent pregnancy were excluded. Patients unable or unwilling to commit to study procedures or those with serious illness that could affect their ability to complete the study were also excluded. Patients showing a clinically relevant improvement in the DME between initial identification and first screening/baseline visit were excluded.

### 2.2. Definition of “Study Eye”

In participants with both eyes eligible, the “study eye” was that with better best-corrected visual acuity (BCVA); if both eyes had equal vision, it was the eye with lowest CRT. 

### 2.3. Primary Outcome

Mean change in BCVA from baseline to week 12 (+7 days). 

### 2.4. Secondary Outcomes

Mean change from baseline to week 12 (+7 days) in CRT; central retinal sensitivity; retinal perfusion; tear production; patient-reported outcomes; proportion of participants with ≥10 and ≥15 Early Treatment Diabetic Retinopathy Study (ETDRS) letter gain; cibinetide antibodies; and adverse events (AEs).

### 2.5. Exploratory Outcomes

Additional data were collected for exploratory analyses to determine potential effects of treatment on the metabolic status of patients, renal function, and on 92 serum inflammatory or reparative proteins. 

### 2.6. Clinical Evaluation and Time Points

Once informed consent was obtained, the following evaluations were undertaken (Figure 1):

Early Treatment Diabetic Retinopathy Study (ETDRS); Best Corrected Visual Acuity (BCVA); Spectral Domain Optical Coherence Tomography (SD-OCT); adverse events (AE); Fundus Fluorescein Angiography (FFA); Columbia Suicide Severity Rating Scale (C-SSRS); EuroQol quality of life questionnaire (EQ-5D-5L); National Eye Institute Visual Function Questionnaire (NEI VFQ-25).

BCVA was obtained using ETDRS charts at 4 m. ETDRS total score was recorded.Mean central 10-degree macular sensitivity (expert test, 10-degree 45-stimulus grid) was determined with the MAIA microperimeter.

BCVA and microperimetry were obtained in both eyes of each participant by experienced optometrists prior to structural evaluations and Schirmer test.

CRT (average thickness in the central 1 mm) was obtained in both eyes by an experienced ophthalmic photographer using the SD-OCT Heidelberg Spectralis. The presence/absence of intraretinal, subretinal and sub-retinal pigment epithelium (RPE) fluid was noted.Retinal perfusion was assessed in both eyes by wide angle fundus fluorescein angiography (WA-FFA), undertaken by an experienced ophthalmic photographer and evaluated qualitatively by an experienced clinician (NL) for the presence/absence/extension of areas of retinal ischaemia at baseline and at week 12.Basal tear production was determined in the study eye using Schirmer’s test, undertaken by a research nurse after a drop of topical anaesthetic was applied. The extent of moisture in mm on the filter paper was recorded.Patient-reported outcomes (PROs) were evaluated by means of the EuroQol 5 Dimensions (EQ-5D-5L) [16] and the National Eye Institute Visual Function Questionnaire (NEI VFQ-25) [17]. Given that one individual with a pre-existing history of depression out of over 200 treated had suicidal ideation while using cibinetide [18], the potential for suicidal ideation was evaluated in this trial using the Columbia Suicide Severity Rating Scale (C-SSRS) [19].Cibinetide antibodies were evaluated in blood samples.Adverse Events (AEs) and Serious Adverse Events (SAEs) were assessed by the Chief Investigator (CI) for causality, seriousness and severity. All SAEs, if there had been any, would have been assessed for expectedness based on the cibinetide Investigator Brochure (IB).

If, at week 12, the macula was fully dry, a visit at week 16 would have been undertaken. Four weeks after completion of treatment patients were phoned to elucidate possible effects experienced after discontinuation of the drug.

To assess metabolic control. HbA1c was determined. HbA1c integrates plasma glucose levels over a period of ~100 days. Glycosylated albumin, a mid-term biomarker of changes in plasma glucose over ~40 days was also determined [20]. Potential effects on renal function were evaluated using urinary albumin/creatinine ratio (ACR) and serum carbamoylated albumin [21]. Finally, 92 serum proteins associated with inflammation, tissue protection and repair were determined (Proseek Multiplex Inflammatory Panel 1; Olink, Uppsala, Sweden). 

### 2.7. Intervention

Participants were instructed to self-administer cibinetide subcutaneously at a dose of 4 mg/daily. A study drug administration guideline was given as additional home reference for self-administration and a study drug diary card to record each dose administered. Treatment continued for 84 days unless discontinued earlier due to withdrawal or >10 letter loss. In the latter event, the study drug was discontinued and standard therapy offered. 

### 2.8. Sample Size and Statistical Analysis

This was a pilot, early phase clinical trial. Descriptive statistics were used for the evaluation of primary and secondary outcomes; no imputation or sensitivity analysis was performed. Mean (+SD) change from baseline was determined for all continuous variables. The percentage of participants with >10 and >15 ETDRS letter gain was planned to be reported. Data obtained at week 16, if available, were to be similarly evaluated. AEs/SAEs were planned to be presented using counts and percentages. All above were part of pre-planned analyses contained in the Statistical Analysis Plan (SAP). Post-hoc exploratory analyses of serum protein profile changes in the concentrations of individual analytes were assessed non-parametrically utilising the Wilcoxon signed rank test. Pre-planned statistical analyses were conducted by C.C. as described in the Statistical Analysis Plan (SAP) using STATA version 12.1 [22]; post-hoc analyses were conducted by M.B.

## 3. Results

Nine participants were enrolled (see Table 1 and Table 2); one experienced a drop of >10 letters at 4 weeks and exited the study to receive standard therapy. Outcomes were evaluated for the remaining eight patients.

Descriptive statistics of BCVA, macular sensitivity, CRT and basal tear production at baseline and week 12 are presented in Table 3. A qualitative evaluation of wide-angle FFA disclosed no changes in retinal perfusion from baseline to week 12.

BCVA did not improve by >5 ETDRS letters in any of the eight study eyes from the eight participants completing the study (i.e., no patient gained ≥10 or ≥15 letters). In seven study eyes, the change in BCVA remained within ± four letters; in one, a 13-letter loss occurred (Table 4). Seven out of eight fellow eyes had DME; one of >400 µm and six of < 400 µm CRT (Table 4). In the latter, five experienced BCVA improvements (three of < five letters, one of eight letters and one of 17 letters); one lost two letters (Table 4). The fellow eye with >400 µm CRT lost one letter (Table 4). 

Improvements in BCVA were more often observed in eyes with thinner retinas (i.e., less CRT) at baseline (Figure 2).

Macular sensitivity improved in three study eyes (by + 0.5, +1.7 and +1.9 dB) and deteriorated in five (by −0.1, −0.6, −1.4, −2.9 and −3.3 dB) (Table 4). Macular sensitivity improved in five of six fellow eyes with CRT of <400µm (by +0.9, +1.1. +1.2, +1.4 and +2 dB) and deteriorated in the other (by −4.1 dB). The fellow eye with >400 µm had a loss of −0.9 dB (Table 4).

In five of eight study eyes, CRT decreased at week 12 (by −4, −7, −30, −66 and −132 µm); in three increased (by +44, +115, +160 µm) (Table 4). CRT was reduced in two of six fellow eyes with CRT of <400µm (by −4 and −9 µm) and increased in four (by +8, +10, +17 and +49 µm) (Table 4). The fellow eye with >400 µm in CRT had a decrease in thickness of 12µm (Table 4). Concordance in the response between eyes (increased or reduced CRT) was observed in all but one patient.

Tear production improved by 1–6 mm in five of eight study eyes, remained the same in one, and decreased in two (Table 4). Improvement was greater in subjects with impaired tear production at baseline (<10 mm) (Table 4). These changes were not reflected on mean tear production values given that in one patient tear production decreased markedly affecting mean levels for the whole group (Table 3).

Mean composite score of the NEI VFQ-25 increased by 2.7 ± 3.1 points, with ocular pain, near activities, and role difficulty domains exhibiting the largest improvements (Table 3). There was an improvement in the EQ-5D 5L visual analogue score of 6.3 units, but there was large variability in the responses (SD 21.5; CI -11.7, 24.2) and the overall score demonstrated no improvement (Table 3).

No Patient Developed Antibodies to Cibinetide.

All patients had elevated HbA1c at baseline [average 74 mmol/L (equivalent to 8.9%)] (Table 1 and Table 2). Treatments for diabetes are shown in Table 2; these did not change for any of the participants during the period of the study. For all eight patients that completed the study mean glucose levels improved (baseline: 75.0 ± 20.9 mmol/L; week 12: 71.8 ± 23.8 mmol/L) (Table 5). Six patients showed improved control and two worsening, including one who had the poorest control at baseline. When glucose control was assessed using serum glycated albumin levels, three patients improved, four worsened and one patient remained unchanged (Table 5).

Renal function, as assessed by the ACR, was abnormal at baseline in four patients (two had macroalbuminuria and two microalbuminuria) (Table 5). All four improved, with the largest changes occurring in individuals with poorer baseline values. No consistent change in carbamoylated albumin was noted. (Table 5).

Serum cytokine data were available at baseline and 12 weeks for six patients. Across all parameters (for further information see [23]), few achieved significant changes from baseline. Mean change from baseline to week 12 was greatest for fibroblast growth factor (FGF) 19, which increased (Table 5). Five patients exhibited moderate to large increases in FGF 19, one showed a very small decrease. FGF 21 levels decreased in four out of six patients (Table 5).

### Safety

All subjects had at least one AE. There were no SAEs. The most frequent AEs include increased triglycerides (5/9), headache (4/9), head cold (4/9), increased glucose (3/9), nausea (2/9), decreased BCVA (2/9), pneumonia (1/9) and staphylococcus aureus of toe (1/9). Since fasting status was not confirmed, triglycerides and glucose elevations, considered AEs because of a change from baseline, may have been postprandial effects. Headache, head cold and nausea were felt to be potentially related to the study drug. Subcutaneous administration was well-tolerated. No patient exhibited suicidal ideation.

## 4. Discussion

In this exploratory trial, cibinetide was safe when administered to patients with DME. Although improvements in mean change in BCVA, central retinal sensitivity, CRT or tear production for the whole group of patients were not seen, improvements in mean NEI VFQ-25 scores were detected.

When evaluating individual participant data, improvements in CRT and tear production were seen in some individuals. Thus, 3/9 (33%) eyes with CRT > 400 microns had a reduction in CRT beyond what could be explained by test–re-test variability [24] and what could be considered the diurnal variability of DME [25]. Although this reduction could potentially be explained by a spontaneous improvement of macular fluid [26], a beneficial effect of cibinetide is possible. In this regard, cibinetide has been shown to have protective effects in endothelial and Muller cells and reduce vascular leakage and retinal edema in experimental rodent models of diabetes [10,11], which could explain these findings.

A linear relationship between improvements in BCVA at 12 weeks and CRT at baseline was observed (Figure 2), suggesting that cibinetide may lead to visual improvement in eyes with milder forms of DME. Moreover, improvements in central retinal sensitivity were observed often in fellow eyes, more often than in study eyes; the former had milder edema than the latter, further suggesting a potential benefit of cibinetide in retinal function in early disease. Given that cibinetide is administered subcutaneously and has the potential to exert an effect in both eyes, evaluating its potential therapeutic effect as an early intervention in eyes with less severe DME, alone or in combination with local therapies such as anti-VEGFs, appears warranted. The effect of cibinetide in retinal function may possibly relate to its apparent neuroprotective effects [11,27,28]. Neuroprotective effects in the retina in experimental diabetic retinopathy models have also been observed with EPO [29,30,31].

Prediabetes and diabetes are associated with a reduction in purely sensory corneal small nerve fibers, which is strongly associated with ocular symptoms of dryness and discomfort [32,33]. Herein, cibinetide led to improved tear production in patients with reduced basal tear formation. NEI VFQ-25 also demonstrated an improvement in ocular pain. In this regard, prior clinical studies of patients with type 2 diabetes [15] and sarcoidosis [18] with painful small fiber neuropathy and reduced corneal nerve fiber abundance showed that a 28-day course of cibinetide increased corneal nerve fibers and reduced ocular pain. Thus, it is likely that the improvements in ocular pain observed in this trial was the result of beneficial effects of cibinetide in the cornea.

ACR is a sensitive assessment of glomerular injury, a hallmark of diabetes-induced injury. All four patients with abnormal ACRs at baseline showed improved indices at week 12. This finding is supported by preclinical data demonstrating efficacy of cibinetide in models relevant to diabetic kidney injury [34]. Carbamoylated albumin concentrations increase in proportion to decreases in glomerular filtration rates. Only one patient had an abnormal concentration at baseline; the reduction at week 12 could indicate a beneficial effect in glomerular filtration rates.

The improved metabolic control with cibinetide observed in some subjects in the trial is consistent with preclinical studies showing increased insulin sensitivity and improved glycemic control in rodent models of diet-induced metabolic syndrome, and with results of a clinical study in patients with type 2 diabetes treated with this peptide, daily, for 28 days [15]. The potential positive signal of improved metabolic control is also supported by the exploratory results of serum multiplex analyses. FGF 19 and FGF 21 have common structural properties and act as classic hormones; administration of FGF 19 or FGF 21 induced weight loss and increased insulin sensitivity in animal models via a direct effect on the central nervous system [35]. FGF 19 is suppressed in obese or diabetic humans and increases following increased insulin sensitivity and weight loss induced by diet or bariatric surgery [36]. In contrast, FGF 21 is increased in human obesity and diabetes, suggesting a FGF 21-resistant condition. The improvement of insulin sensitivity and glycemic control in humans results in lowering of serum FGF 21 concentrations [36]. Cibinetide reduced elevated FGF 21 in both serum and muscle of diet-induced insulin resistance and hyperglycemic mice [13]. The reduced levels of FGF 19 and increased levels of FGF 21 observed at baseline in patients enrolled in this trial and the trend for reversal following cibinetide could be related to improved metabolic control, which would be consistent with changes observed in HbA1c.

One potential confounder could be the fixed 4-mg dose used. IRR signalling requires systemic concentrations of cibinetide ≥1 nM. In non-obese subjects, the peak concentration of a 4 mg subcutaneous dose in pharmacokinetic studies was ~1.8 nM [37]. It is possible that peak levels of cibinetide in obese participants may have been less than the minimum activating concentration. In this regard, the patient that terminated the study early had a BMI of 64.8.

The limitations of this trial include its pilot nature, very small number of patients included, lack of control group and very short duration of follow-up. However, the trial demonstrated, for the first time, that cibinetide is safer when administered for longer periods than those previously reported and may have favourable effects in patients with diabetes and DME. The strengths include its rigorous methodology and meticulous functional and structural assessments combined with an in-depth systemic evaluation. Positive potential signals, including improved patient-reported visual functional outcomes, ocular pain, and, in some patients, clearance of macular fluid, increased tear production and improved metabolic and kidney function, warrant further investigation.

## Figures and Tables

**Figure 1 jcm-09-02225-f001:**
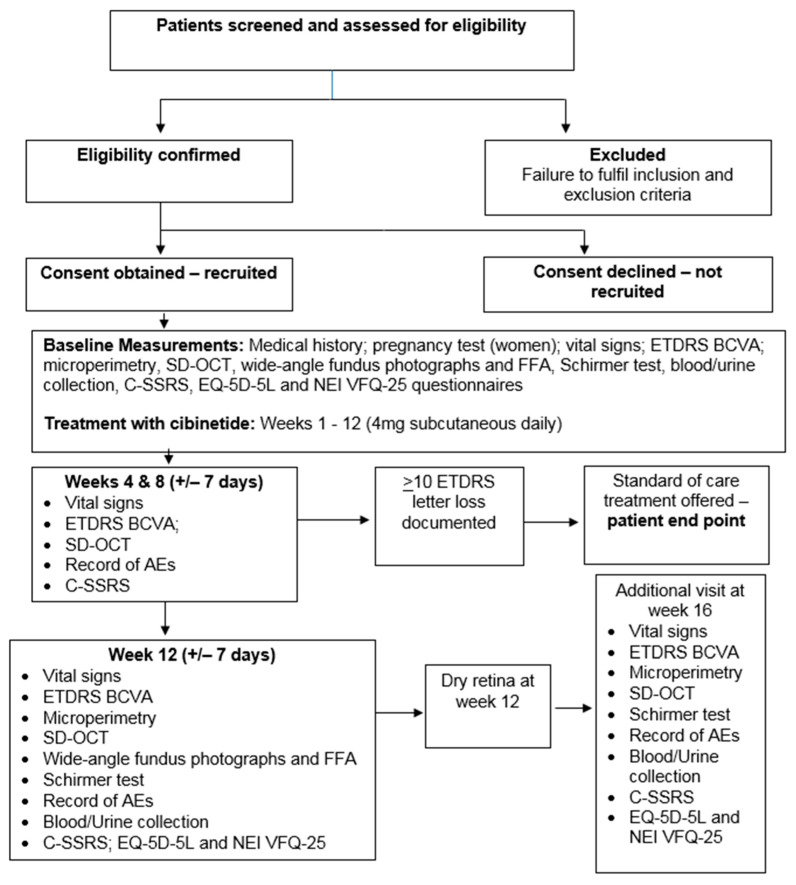
Study flow chart for cibinetide in diabetic macular edema (DME) phase 2 clinical trial.

**Figure 2 jcm-09-02225-f002:**
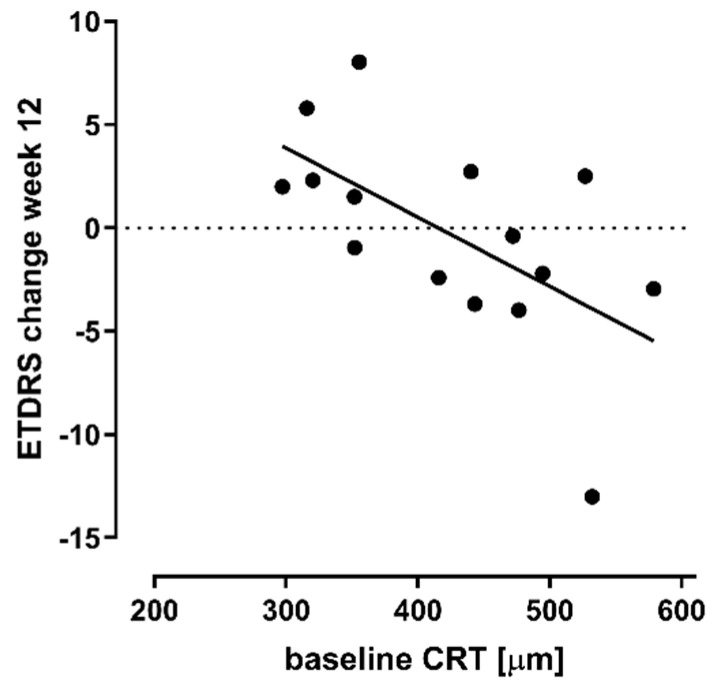
Early Treatment Diabetic Retinopathy Study (ETDRS) letter change from baseline to week 12 plotted against values of baseline central subfield retinal thickness (CRT). Eyes with less severe diabetic macular edema and thinner retinas (less CRT) at baseline experienced more gain in vision than those with thicker retinas. (Regression line: ETDRS change = −0.033 × (baseline CRT) + 13.9; r2 = 0.36; *p* = 0.017).

**Table 1 jcm-09-02225-t001:** Baseline characteristics of participants.

Characteristics		Baseline *n* = 9(100%)	Baseline * *n* = 8(88.9%)
Gender	Male	5 (55.6%)	5 (62.5%)
	Female	4 (44.4%)	3 (37.5%)
Study Eye	Left	2 (22.2%)	2 (25.0%)
	Right	7 (77.8%)	6 (75.0%)
Fellow Eye with DME present	≥400 microns	2 (22.2%)	1 (12.5%)
	<400 microns	6 (66.7%)	6 (75.0%)
Type of diabetes	Type 1	1 (11.1%)	1 (12.5%)
	Type 2	8 (88.9%)	7 (87.5%)
Duration of diabetes (years, mean ± SD and median [IQR])		14.4 ± 8.5 16.1 [9.8, 18.0]	15.0 ± 8.9 16.7 [8.8, 20.5]
Age (years, mean ± SD)		57.6 ± 13.9	57.6 ± 14.9
Weight (kg, mean ± SD)		95.5 ± 20.5	90.6 ± 15.3
Height (m, mean ± SD)		1.6 ± 0.12	1.7 ± 0.1
Body Mass Index (kg/m2, mean ± SD)		37.18 ± 12.1	33.6 ± 5.9
HbA1c (mmol/L, mean ± SD)		74.0 ± 19.7	75 ± 20.9

* = Baseline characteristics of the eight patients that completed the trial to week 12.

**Table 2 jcm-09-02225-t002:** Baseline characteristics of participants.

	Baseline								
Patient	A	B	C	D	E	F	G	H	I
Gender	Female	Male	Male	Male	Male	Male	Female	Female	Female
Study Eye	Right	Left	Right	Left	Right	Right	Right	Right	Right
Type of diabetes	2	1	2	2	2	2	2	2	2
Duration of diabetes (years)	2.7	18	23	14.9	17.3	16.1	9.8	26.5	1.1
Age (years)	74.5	31.8	43.7	50.8	58.5	59.7	57.7	72	69.5
Weight (kg)	74.2	81.5	118	92.2	102	94.4	134.6	91.2	71
Height (m)	1.57	1.75	1.65	1.66	1.72	1.76	1.43	1.49	1.56
BMI (kg/m2)	30.1	26.6	43.3	33.5	34.5	30.5	65.8	41.1	29.2
HbA1c (mmol/L)	63	83	56	117	88	74	66	65	54
Diabetes Medications	Metformin	Novorapid	Sitagliptin	Metformin	Dapagliflozin	Novorapid insulin	Metformin	Metformin	Gliclazide
		Lantus Solostar	Gliclazide	Novo Rapid	Novomix 30	Levemir	Novorapid Flexpen	Novomix 30	Metformin
				Levemir	Metformin		Liraglutide		
					Novomix 30 (PM)		Lantus		

**Table 3 jcm-09-02225-t003:** Descriptive statistics for outcome measures at baseline and at week 12.

	Baseline	Baseline *	Week 12 *	Change *
	*n =* 9 (100%)	*n =* 8 (88.9%)	*n =* 8 (88.9%)	*n =* 8 (88.9%)
Best corrected distance visual acuity (mean ± SD and 95% CI)	68.1± 7.9 (62.0, 74.2)	69.1 ± 7.8 (62.6, 75.6)	66.3 ± 9.5 (58.3, 74.2)	−2.9 ± 5.0 (−7.1, 1.3)
Central subfield thickness (microns, mean ± SD and 95% CI)	490.3 ± 61.9 (442.8, 537.9)	488.5 ± 65.9 (433.4, 543.6)	498.5 ± 127.1 (392.2, 604.8)	10.0 ± 94.6 (−69.1, 89.1)
Central retinal sensitivity (dB, mean ± SD and 95% CI)	23.3 ± 2.2 (21.5, 25.0)	23.7 ± 1.9 (22.1, 25.3)	23.2 ± 2.3 (21.3, 25.1)	−0.53 ± 1.9 (−2.1, 1.1)
Tear production (mm, mean ± SD and 95% CI)	13.4 ± 8.8 (6.7, 20.2)	13.9 ± 9.3 (6.1, 21.7)	13.8 ± 2.4 (11.8, 15.7)	−0.13 ± 7.7 (−6.6, 6.3)
VFQ-25 Composite Score (mean ± SD and 95% CI)	79.0 ± 20.1 (63.6, 94.5)	84.8 ± 11.3 (75.4, 94.2)	87.5 ± 6.9 (81.8, 93.2)	2.7 ± 3.1 (−4.5, 10.0)
EQ-5D-5L Index (mean ± SD and 95% CI) EQ-5D 5L Visual Analogue Score (mean ± SD and 95% CI)	0.7 ± 0.4 (0.5, 1.0) 68.9 ± 21.3 (52.5, 85.3)	0.8 ± 0.2 (0.7, 1.0) 68.8 ± 22.8 (49.7, 87.8)	0.8 ± 0.3 (0.6, 1.0) 75.0 ± 12.8 (64.3, 85.7)	−0.03 ± 0.2 (−0.2, 0.1) 6.3 ± 21.5 (−11.7, 24.2)

* = data evaluated for the eight patients that completed the trial to week 12.

**Table 4 jcm-09-02225-t004:** Individual visual and patient-reported outcomes for all patients that completed the study (*n =* 8) at baseline and at week 12.

		Best Corrected Distance Visual Acuity			Central Subfield Thickness (Microns)			Central Retinal Sensitivity (dB)			Tear Production (mm)			EQ5D-5L Index Score			VFQ Composite Score		
Patient	Eye	Baseline	Week 12	Change	Baseline	Week 12	Change	Baseline	Week 12	Change	Baseline	Week 12	Change	Baseline	Week 12	Change	Baseline	Week 12	Change
**A**	**Study Eye**	73	69	−4	602	646	44	22.7	24.4	1.7	10	16	6	0.61	0.29	−0.32	87.4	89.0	1.7
	**Non Study Eye**	79	77	−2	375	424	49	24.1	25.5	1.4	-	-	-						
**B**	**Study Eye**	68	70	2	538	653	115	25.6	22.7	−2.9	35	17	−18	0.94	1.00	0.06	87.7	80.6	−7.1
	**Non Study Eye**	79	80	1	363	380	17	26.5	27.6	1.1	-	-	-						
**C**	**Study Eye**	59	56	−3	428	588	160	23.9	20.6	−3.3	11	14	3	0.95	0.95	0.00	86.6	95.9	9.3
	**Non Study Eye**	85	88	3	305	315	10	25.4	21.3	−4.1	-	-	-						
**D**	**Study Eye**	84	81	−3	429	425	−4	23.7	24.2	0.5	6	11	5	1.00	1.00	0.00	85.1	81.6	−3.5
	**Non Study Eye**	69	68	−1	485	473	−12	23.1	22.2	−0.9	-	-	-						
**E**	**Study Eye**	72	68	−4	477	470	−7	23.6	22.2	−1.4	17	15	−2	0.89	0.84	−0.05	98.2	91.4	−6.8
	**Non Study Eye**	70	78	8	356	364	8	21.1	22.3	1.2	-	-	-						
**F**	**Study Eye**	61	59	−2	490	424	−66	27	26.9	−0.1	6	10	4	0.65	0.89	0.24	73.3	82.9	9.7
	**Non Study Eye**	78	84	6	311	304	−725	28	26.9	−1.1	-	-	-						
**H**	**Study Eye**	70	74	4	412	280	−132	22.5	24.4	1.9	14	14	0	0.63	0.48	−0.15	64.0	81.2	17.2
	**41**	41	58	17	239	230	−9	21.9	22.8	0.9	-	-	-						
**I**	**Study Eye**	66	53	−13	532	502	−30	20.6	20	−0.6	12	13	1	1.00	1.00	0.00	95.9	97.4	1.5
	**Non Study Eye**	72	74	2	297	293	−4	22.8	24.8	2	-	-	-						

Note: Patient labelled as G experienced a 10 letter drop at month 4 in the study eye; data from this patient are not presented herein.

**Table 5 jcm-09-02225-t005:** Individual systemic parameters for all patients that completed the study (*n =* 8) at baseline and at week 12.

	BMI (kg/m2)		Systolic Blood pressure (mmHg)		Diastolic Blood Pressure		HbA1c (mmol/L)		HDL (mmol/L)		LDL (mmol/L)		Triglycerides (mmol/L)		Albumin Creatinine Ratio		Carbamoylated Albumin		Glycated Albumin		FGF-19		FGF-21		Albumin (g/L)		Estimated GFR (mL/min)	
Patient	Baseline	Week 12	Baseline	Week 12	Baseline	Week 12	Baseline	Week 12	Baseline	Week 12	Baseline	Week 12	Baseline	Week 12	Baseline	Week 12	Baseline	Week 12	Baseline	Week 12	Baseline	Week 12	Baseline	Week 12	Baseline	Week 12	Baseline	Week 12
A	30.1	-	167	144	72	70	63	53	1.4	1.4	1.4	1	1.33	1.82	5.5	3.41	2.4	2.6	18.1	17.6	7.2	9.7	5.5	4.6	48	47	>60	>60
B	26.6	-	129	124	76	80	83	75	1.3	1.2	3.5	2.5	0.68	0.53	46.5	40.64	2.8	3.1	22.1	20.3	6.6	6.4	2.8	5.6	43	45	>60	>60
C	43.3	-	131	160	71	73	56	50	1	0.9	1.6	1.5	2.71	2.9	220.65	195.79	4.9	4.8	16.9	14.7	7.8	-	4.1	-	36	40	41	38
D	33.5	-	126	128	81	81	117	123	1.1	1	1.7	1.3	1.43	2.32	1.6	2.14	2.5	2.3	23.5	26.0	7.8	9.0	9.3	6.8	46	46	>60	>60
E	34.5	-	142	125	79	66	88	85	0.8	0.9	2.7	2.5	2.52	2.62	4.52	3.59	1.7	1.7	17.4	18.5	-	7.7	-	6.1	43	47	>60	>60
F	30.5	-	152	126	56	62	74	68	0.9	1	1.3	1.7	1.49	1.41	1.38	0.73	1.8	2.0	23.9	24.8	6.1	10.6	5.7	2.6	40	40	>60	>60
H	41.1	-	150	127	71	64	65	64	1.3	1.4	1.5	1.5	2.49	2.6	3.1	2.71	2.5	3.1	18.9	21.2	6.8	7.6	6.9	6.0	43	43	>60	>60
I	29.2	-	136	122	74	71	54	56	1.6	1.5	3.3	3	1.91	1.91	3.04	3.1	2.6	2.7	14.5	14.3	6.8	7.2	2.2	5.9	47	45	>60	>60

Note: Patient labelled as G experienced a 10 letter drop at month 4 in the study eye; data from this patient are not presented herein. Normal values for local laboratory where analysis took place are as follows: HbA1c <53 mmol/L; HDL >1.0 mmol/L; LDL <3.0 mmol/L; triglycerides <2.2 mmol/L; albumin 35–50 g/L; estimated Glomerular Filtration Rate (eGFR) >60 mL/min.

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
