# Peer review of "A Phase 2 Clinical Trial on the Use of Cibinetide for the Treatment of Diabetic Macular Edema"

_jcm, 2020, doi:10.3390/jcm9072225_

Round 1

Reviewer 1 Report

This manuscript by Lois et al. reports on an exploratory trial in which 8 patients with diabetic macular edema have daily self-administered cibinetide 4mg/day subcutaneously for 12 weeks. The main outcome was change in best-corrected visual acuity. Cibinetide is a synthetic 11 amino acid peptide derived from erythropoietin. It supposedly recapitulates the beneficial effects of EPO, but without potential side effects from haematopoiesis. In experimental models, cibinetide reduces macular edema and neuroglial degeneration.

In this pilot study, best-corrected visual acuity hasn’t improved. The secondary outcomes (central retinal thickness, retinal sensitivity, tear production) didn’t change significantly. No serious adverse evens were seen. There was a positive trend in the patient reported NEI VFQ-25 (visual function) and EQ-5D 5L visual analogue score, possibly related to improved tear production, increased corneal nerve fibers. Hence, the authors conclude that positive signals in patient-reported visual function and ocular pain warrant further investigations.

While the manuscript is clearly structured and neatly written, findings are somewhat overstated in the conclusions (lines 43-44 on page 1). None of the positive trends was statistically significant.

Line 86 on page 2: brackets missing for reference# 10

Author Response

While the manuscript is clearly structured and neatly written, findings are somewhat overstated in the conclusions (lines 43-44 on page 1). None of the positive trends was statistically significant.

We agree with Reviewer 1 in that changes were not statistically significant. We have modified, thus, the Conclusion statement, which now reads:

“Cibinetide 12-week course was safe. Improvements in NEI VFQ-25 scores, CRT, tear production, diabetic control and albuminuria, observed in some participants, warrant further investigation”

Line 86 on page 2: brackets missing for reference# 10

Thank you very much for pointing out this error. We have corrected it in the revised version of the manuscript.

Reviewer 2 Report

The article 'A phase 2 clinical trial on the use of cibinetide for the treatment of diabetic macular edema' is well written and structured. This will help the scientific community dealing with vision science mainly those who are involved in diabetic macular edema (DME) research. The most important aspect of this article is the evaluation and effects of cibinetide in (DME) that will eventually lead to develop a better drug in future compared to erythropoietin (EPO) which has side effects.

Author Response

We thank Reviewer 2 for her/his comments about our manuscript.  It seems that Reviewer 2 is content with our work and how it is presented and did not raise any questions for us.

Reviewer 3 Report

Dear Authors, this is an interesting paper on the effect of cibinetidine for the treatment of macular edema. I recommend this paper to be suitable for publication after major changes. Please find below my recommendations:

  1. "patients with type 1 and type 2 diabetes" were included in the study.  Since you mentioned that the patient that finished the study earlier had BMI 64.8, please include in the table the range of the BMI, weight and median. Reformulate Table 1 including the diabetes duration and baseline treatment for diabetes. 
  2. Please explain why did you use the only the carbamylated albumin for the kidney function evaluation? The authors of the cited paper-21- state in their discussion that in humans the role of carbamylated albumin is unknown, especially if there is a previous alteration of the GFR. Please include data on the eGFR at baseline and at the end of the study. Also, in order to correctly interpret and evaluate the levels of carbamylated albumin, glycated albumin and HbA1c you need to refer to the Hb and albumin levels. Please include these data in Table 4. 
  3. Table 4: Include data on weight in each patient, state the normal values of your local lab on the biochemical parameters that are explained in this table. Include data on eGFR. 
  4. Table 2: data is scarce compared to the complexity of the exploration that were performed. Please include data on gaze fixation evaluated by microperimetry, retinal perfusion, etc.
  5. Discussion: the main part of the discussion is centered on the metabolic control. Nevertheless, this study was not designed for the evaluation of the metabolic changes induced by  cibinetide. The statement between line 332 and 337 is not correct. The improvement that is explained in the ACR at week 12 is not visible from the Table 4. This conclusion can not be withdrawn from this study and the presented data. 
  6. Discussion: paragraph between  lines 338-353 is not correct. During the study, 2 patients out of 8 worsened the HbA1C levels after 12 weeks and 4 patients worsened the glycated albumin, which is a more sensitive parameters for short periods of time. Authors can not draw the conclusion of metabolic improvement from this study. Additionally, no data is presented regarding the baseline treatment of diabetes and any possible changes in this medication during the study. 
  7. This study had as primary and secondary outcomes only ophthalmological parameters. Nevertheless, the main part of the discussion is on metabolic changes, and conclusions that cannot be withdrawn from this study. I recommend to center the discussion on the ophthalmological findings, giving some possible hypothesis and explanations and future perspective. The effect on the metabolic status only can be suggested but can not be supported by the results of this study. 

As the authors stated in the end of the manuscript, the study demonstrates for the first time that cibinetidine is safe and may have favourable effects in patients with diabetes and DME. Complete ophthalmological evaluation was performed, by little data is shown.  I recommend to leave the metabolic findings in a secondary plan, and detail and comment more on ophthalmological findings.

After major changes I would be happy to reevaluate the paper.

Author Response

REVIEWER 3

Dear Authors, this is an interesting paper on the effect of cibinetidine for the treatment of macular edema. I recommend this paper to be suitable for publication after major changes. Please find below my recommendations:

  1. "patients with type 1 and type 2 diabetes" were included in the study.  Since you mentioned that the patient that finished the study earlier had BMI 64.8, please include in the table the range of the BMI, weight and median. Reformulate Table 1 including the diabetes duration and baseline treatment for diabetes. 

Thank you very much for the suggestion.  Table 1 has already information on mean and SD for weight, height and body mass index (BMI) for all patients in the study.  Thus, we have added another table (we have now labelled table 1 - 1A, and added table 1B) with each individual patient and provided information on type of diabetes, duration of diabetes, weight, height, BMI, and treatment for diabetes, so all this information will be in the same table and for each patient.

We hope this will address the issue raised by Reviewer 3. 

  1. Please explain why did you use the only the carbamylated albumin for the kidney function evaluation? The authors of the cited paper-21- state in their discussion that in humans the role of carbamylated albumin is unknown, especially if there is a previous alteration of the GFR. Please include data on the eGFR at baseline and at the end of the study. Also, in order to correctly interpret and evaluate the levels of carbamylated albumin, glycated albumin and HbA1c you need to refer to the Hb and albumin levels. Please include these data in Table 4. 

We did not use only carbamylated albumin to assess kidney function.  We obtained urine albumin to creatinine ratio (ACR), as stated in the manuscript (Methods section, lines 189 and 190).  Values of ACR for each participant at baseline and at week 12 are presented in table 4. 

We did obtain also eGFR values at baseline and at the end of the study for all participants; these have been added now, as requested by the Reviewer, to table 4.

We did record albumin levels in the participants at baseline and at the 12 week follow-up.  We have added these values to table 4 also, as suggested by Reviewer 3.  However, blood cell count and haemoglobin values were only recorded if outside normal range (they were recorded to evaluate any possible adverse events and serious adverse events).  As a result, values of haemoglobin are only available for those participants and in those visits in which they were abnormal.  Thus, unfortunately, we will not be able to add these data to the table (we do not think it would be helpful to add the data we have for some of the patients/study visits given that we do not have them for all patients/visits).

  1. Table 4: Include data on weight in each patient, state the normal values of your local lab on the biochemical parameters that are explained in this table. Include data on eGFR. 

As per the above comments, we have added a new table, Table 1B, which contains information on weight, height and BMI (as well as other additional information) for each participant.

We have also added the normal values for our laboratory, as requested by the Reviewer, to table 4.

As stated above, we have also added eGFR values to table 4.

  1. Table 2: data is scarce compared to the complexity of the exploration that were performed. Please include data on gaze fixation evaluated by microperimetry, retinal perfusion, etc.

We did not plan in our protocol to evaluate fixation on microperimetry in this study, but only to assess retinal sensitivity (see Methods section, Secondary Outcomes, lines 122-126).  We would prefer not to add data to the manuscript that is outside the protocol and which we were not planning to report. However, if the Reviewer considers the addition of this information essential we would oblige, as we do have information on fixation for the participants. 

We had planned to measure areas of capillary dropout quantitatively if possible.  However, this could not be done reliably.  Thus, only the qualitative assessment was presented (see Results section, lines 221, 222).

  1. Discussion: the main part of the discussion is centered on the metabolic control. Nevertheless, this study was not designed for the evaluation of the metabolic changes induced by cibinetide. The statement between line 332 and 337 is not correct. The improvement that is explained in the ACR at week 12 is not visible from the Table 4. This conclusion cannot be withdrawn from this study and the presented data. 

We had stated in the Discussion section (lines 332-337): “ACR is a sensitive assessment of glomerular injury, a hallmark of diabetes-induced injury.  All four patients with abnormal ACRs at baseline showed improved indices at week 12.  This finding is supported by preclinical data demonstrating efficacy of cibinetide in models relevant to diabetic kidney injury [27].  Carbamoylated albumin concentrations increase in proportion to decreases in glomerular filtration rates. Only one patient had an abnormal concentration at baseline; the reduction at week 12 could indicate a beneficial effect in glomerular filtration rates.”

We do think our statement is correct.  In table 4, patients A, B, C and E had abnormal ACR (5.5, 46.5, 220 and 4.52, respectively) and these values improved at week 12 (to 3.41, 40.64, 195.79 and 3.59, respectively).

We have not changed or added further information to the manuscript about this issue.

  1. Discussion: paragraph between  lines 338-353 is not correct. During the study, 2 patients out of 8 worsened the HbA1C levels after 12 weeks and 4 patients worsened the glycated albumin, which is a more sensitive parameters for short periods of time. Authors can not draw the conclusion of metabolic improvement from this study. Additionally, no data is presented regarding the baseline treatment of diabetes and any possible changes in this medication during the study. 

The statement in the Discussion section of our manuscript in lines 338-353 read as follows:

The observed improved metabolic control with cibinetide is consistent with preclinical studies showing increased insulin sensitivity and improved glycemic control in rodent models of diet-induced metabolic syndrome,[15] and with results of a clinical study in patients with type 2 diabetes treated with this peptide, daily, for 28 days [15].  The potential positive signal of improved metabolic control is also supported by the exploratory results of serum multiplex analyses. FGF 19 and FGF 21 have common structural properties and act as classic hormones; administration of FGF 19 or FGF 21 induced weight loss and increased insulin sensitivity in animal models via a direct effect on the central nervous system [28].  FGF 19 is suppressed in obese or diabetic humans and increases following increased insulin sensitivity and weight loss induced by diet or bariatric surgery [29].  In contrast, FGF 21 is increased in human obesity and diabetes, suggesting a FGF 21-resistant condition. Improvement of insulin sensitivity and glycemic control in humans results in lowering of serum FGF 21 concentrations [29].  Cibinetide reduced elevated FGF 21 in both serum and muscle of diet-induced insulin resistance and hyperglycemic mice [13].  The reduced levels of FGF 19 and increased levels of FGF 21 observed at baseline in patients enrolled in this trial and the trend for reversal following cibinetide are likely related to improved metabolic control, which is consistent with changes observed in HbA1c”

We have modified it by “softening” our statement, as follows:

 “The improved metabolic control with cibinetide observed in some subjects in the trial is consistent…”  “The reduced levels of FGF 19 and increased levels of FGF 21 observed at baseline in patients enrolled in this trial and the trend for reversal following cibinetide could be related to improved metabolic control, which would be consistent with changes observed in HbA1c.”

We hope this is acceptable for the Reviewer.

We have added now to the manuscript a table with the medications for the diabetes the participants were on at baseline (Table 1B). These medications did not change during the period of the study.  We have added information on this matter to the manuscript too.

  1. This study had as primary and secondary outcomes only ophthalmological parameters. Nevertheless, the main part of the discussion is on metabolic changes, and conclusions that cannot be withdrawn from this study. I recommend to center the discussion on the ophthalmological findings, giving some possible hypothesis and explanations and future perspective. The effect on the metabolic status only can be suggested but can not be supported by the results of this study.  As the authors stated in the end of the manuscript, the study demonstrates for the first time that cibinetidine is safe and may have favourable effects in patients with diabetes and DME. Complete ophthalmological evaluation was performed, by little data is shown.  I recommend to leave the metabolic findings in a secondary plan, and detail and comment more on ophthalmological findings.

We thank the Reviewer for this comment and suggestion.  We have expanded the Discussion on the ocular findings (and, subsequently, needed to add few more references to the manuscript).  However, we would wish to maintain the discussion we have on the systemic effects of cibinetide as we consider these important too (although we have modified our comments on the latter, as per Reviewer’s 3 suggestions).